# Dual Resistance to Maribavir and Ganciclovir in Transplant Recipients

**DOI:** 10.3390/v17030421

**Published:** 2025-03-14

**Authors:** Steven B. Kleiboeker

**Affiliations:** Eurofins Viracor Clinical Diagnostics, 18000 West 99th Street, Lenexa, KS 66219, USA; steve.kleiboeker@viracor.eurofinsus.com; Tel.: +1-816-835-3436; Fax: +1-913-408-6240

**Keywords:** cytomegalovirus, resistance, maribavir, ganciclovir

## Abstract

Background: Human cytomegalovirus (CMV) remains an important pathogen, especially for immunocompromised patients such as solid organ and hematopoietic stem cell recipients. Viral genomic mutations conferring drug resistance are an important impediment to effective CMV management and frequently lead to use of alternative antiviral drugs to treat CMV disease. Methods: Results from 1459 de-identified patient samples with both UL54 and UL97 sequencing results were analyzed for ganciclovir (GCV) and maribavir (MBV) resistance mutations. Genomic sequencing was performed by the Sanger method and resistance mutations were identified by comparison to CMV reference strain AD169. Results: Ganciclovir resistance was identified in 379 of 1459 (25.98%) of the samples tested, with most resistance-conferring mutations present in viral gene UL97. A total of 121 of 1459 (8.29%) samples had MBV resistance mutations, and 84 (69.42%) of the 121 samples with MBV resistance also had GCV resistance mutations. Of the 84 samples with resistance to both MBV and GCV, 35 (41.67%) had a single UL97 mutation conferring resistance to both drugs, either C480F or F342Y. The overall prevalence of C480F was increased relative to an earlier analysis of samples from this reference laboratory. Conclusions: Although a high prevalence of CMV resistance mutations was identified, this must be taken in the context of healthcare providers submitting samples from patients with suspected CMV resistance. Most MBV-resistant samples were also resistant to GCV, suggesting that use of MBV as an alternative to GCV may benefit from genotypic resistance testing to achieve the effective control of CMV disease.

## 1. Introduction

Human cytomegalovirus (CMV) is an important pathogen of immunocompromised patients such as solid organ transplantation (SOT) or allogeneic hematopoietic cell transplantation (HCT) recipients, and antiviral drugs are important for patient management. Ganciclovir (GCV) and the prodrug valganciclovir are frequently used for treatment and prophylaxis for CMV. Maribavir (MBV) is newer drug that was recently approved for the treatment of refractory CMV infections with or without demonstrated genotypic resistance mutations for other antiviral drugs.

Drug resistance is a complication for patients receiving CMV antiviral prophylaxis or treatment, and genotypic resistance is well documented for all currently approved anti-CMV drugs [1]. Resistance should be suspected in SOT recipients with unchanged or increasing CMV viral loads or unresolved CMV disease after ≥2 weeks of appropriately dosed treatment or >6 weeks of GCV exposure [2,3]. For HCT recipients on anti-CMV therapy, testing is recommended when the viral load decreases less than 10-fold after >2 weeks [4,5,6]. Risk factors for antiviral resistance include prolonged treatment with viral replication while on the drug, subtherapeutic drug levels, high levels of immunosuppression and lack of CMV immunity [7,8,9,10], all of which may be commonly observed in SOT and HCT recipients. Patient samples with documented resistance mutations are most commonly resistant to a single CMV antiviral drug, although resistance to more than one drug has been reported [1,11]. Resistance to multiple CMV antiviral drugs (e.g., dual resistance) may be due to a single mutation conferring resistance to more than one drug or to the presence of multiple resistance mutations, each conferring resistance to different antiviral drugs.

In this study, de-identified results from samples submitted to a reference laboratory for GCV and MBV genotypic resistance testing were analyzed. Testing requests were based on suspected resistance, the history of antiviral drugs used and/or the need for different treatment or prophylaxis options. Due to de-identification requirements, the type of transplant, disease condition and antiviral treatment history were not available. Results demonstrated that GCV resistance was frequently identified and that a majority of MBV-resistant samples also have mutations conferring resistance to GCV.

## 2. Methods

### 2.1. Samples

A total of 1459 patient samples submitted by healthcare providers to Eurofins Viracor’s reference laboratory for CMV genotypic resistance testing were fully de-identified prior to retrospective analysis. Analysis was performed for GCV resistance mutations in UL54 and UL97 and MBV resistance mutations in UL97. Only samples with full sequencing of both UL54 and UL97 regions of interest were included for analysis. The minimum viral load requested in samples submitted for genotypic resistance testing was 1000 IU/mL.

### 2.2. Sequencing and Analysis

Total nucleic acid was extracted from plasma using Qiagen DNA Blood Mini kits (Qiagen, cat. no. 51104; Qiagen, Germantown, MD, USA), followed by PCR amplification with primers previously reported [12,13] using proof-reading FailSafe Taq polymerase (Epicentre cat. no. FSP995D and FSE5101K, ThermoFisher Scientific, Carlsbad, CA, USA) to generate overlapping PCR products for viral UL54 and UL97 genes. PCR amplicon purification was performed with Qiagen QIAquick PCR Purification Kit (Qiagen, cat. no. 28183). Bi-directional Sanger sequencing was performed using the BigDye^®^ Terminator v3.1 Cycle Sequencing Kit (Life Technologies, cat. no. 4337456, ThermoFisher Scientific, Carlsbad, CA, USA) run on an ABI 3730xl capillary sequencer with ABI POP-7 Polymer (Life Technologies, cat. no. 4363929). Acceptance criteria were trace scores ≥30, quality values ≥ 20, 95% bi-directional coverage, and control sequence matching 100% with the reference sequence in the reportable range. Sequences were analyzed with ABI’s SeqScape 3 software v3.0. Translated nucleotide sequences were compared to the reference CMV strain AD169 amino acid sequence to identify resistance-conferring mutations. Sequencing results encompass UL97 codons 335–613 and UL54 codons 290–989.

## 3. Results

### 3.1. Frequency of Resistance Mutations by Viral Gene and Anti-Viral Drug

Genotypic analysis of viral genes UL54 and UL97 was performed on 1459 patient samples submitted for CMV antiviral resistance mutation testing. Samples included in this analysis had full sequencing of both UL54 and UL97 over the codons of interest for GCV and MBV drug resistance mutations. The resistance mutations identified and the frequency of detection are shown in Table 1 for each viral gene and antiviral drug. Resistance mutations were most commonly identified in UL97 with 24.19% (353 of 1459) and 8.29% (121 of 1459) of samples positive for GCV and MBV resistance mutations, respectively (Table 2). Ganciclovir resistance mutations in UL54 were less commonly identified, with 4.25% (62 of 1459) of samples positive. Ganciclovir resistance mutations were present in both UL54 and UL97 for 2.47% (36 of 1459) samples analyzed.

For UL54, the most common GCV resistance mutations were D413E, L501I, T503I, P522S and A987G (Table 1). For UL97, the GCV resistance mutations M460I, M460V, C480F, H520Q, C592G, A594V, L595F, L595S and C603W were most commonly identified and found in 84.42% (298 of 353) of samples with GCV resistance in UL97, either alone or in combination with other GCV resistance mutations. For UL97 MBV resistance mutations, T409M, H411Y and C480F were most commonly identified and present in 92.56% (112 of 121) of samples with resistance, either alone or in combination with other MBV resistance mutations. Mutations F342Y and C480F, which confer cross-resistance to GCV and MBV, were present in 36.36% (44 of 121) of samples with MBV resistance and in 12.46% (44 of 353) of samples with GCV resistance. Samples with more than one resistance mutation in the same gene and for the same antiviral drug were also observed in UL54 and UL97 and although generally at a low frequency (Table 1). Resistance to GCV in UL97 had the highest number of multiple mutations in individual samples, representing 7.65% (27 of 353) of the samples with mutations detected.

### 3.2. Frequency of Single and Dual Resistance by Patient Sample

Of the 1459 patient samples analyzed, 416 (28.51%) were resistant to GCV and/or MBV. A total of 379 samples (25.89%) were resistant to GCV, 121 samples (8.29%) were resistant to MBV, and 84 samples (5.76%) were resistant to both MBV and GCV (Table 2). Resistance mutations for GCV were predominantly identified in the viral UL97 gene (317 of 379 resistant samples). Ganciclovir resistance mutations were identified only in the UL54 gene for 26 samples.

Of the 84 samples with dual resistance to MBV and GCV, 35 (41.67%) had a single UL97 mutation conferring cross-resistance to both drugs, either C480F or F342Y. The remaining dual-resistant samples had GCV mutations primarily in the UL97 gene (39 of 84) while 4 of 84 samples had a GCV mutation in UL54 and 6 of 84 samples had GCV mutations in both UL97 and UL54. Fisher’s exact test was applied to assess an association between GCV and MBV resistance and it was found to be highly significant (*p* < 0.001).

## 4. Discussion

The prevalence of resistance to anti-CMV drugs varies depending on the drug, study and patient population. Resistance rates for kidney recipients on GCV prophylaxis were reported to be <2% [14] while resistance rates for patients treated with GCV and valganciclovir were 2.3% and 3.6%, respectively [14]. Higher risk SOT recipients have higher rates of resistance, ranging from 5.2 to 12.5% [15,16,17,18]; and a resistance rate of 18% was reported in lung recipients [19]. For intestinal and multivisceral SOT recipients with CMV infection, 31.3% had resistance mutations [20]. As in SOT recipients, HCT recipients in higher risk categories have higher rates of resistance to anti-CMV drugs. Haploidentical recipients receiving preemptive therapy experienced a resistance rate of 14.5% resistance (Shmueli et al., 2015) [21]. In HCT patients treated pre-emptively with GCV, a resistance rate of 7.7% was reported [22]. However, CMV resistance is relatively uncommon after conventional HCT [23].

In the present study, 28.51% of samples analyzed had one or more GCV and/or MBV resistance mutations. This prevalence of resistance is generally higher than rates reported in clinical studies since physicians are submitting samples from patients who are either failing to respond or responding sub-optimally anti-CMV therapy. A similar overall level of resistance was detected in a previous study from the same reference laboratory [11,23], indicating that the prevalence of CMV drug resistance is relatively stable over the past decade or more, although new resistance mutations have been regularly described.

Previous analysis of samples submitted in an earlier time frame to the same reference laboratory revealed similar trends of MBV and GCV resistance [23]. Notably, in the earlier study dual resistance to MBV and GCV was present in 69.4% (86 of 124) samples with MBV resistance, a prevalence nearly identical to the present study. However, C480F was identified in 2.60% of the samples sequenced in the present study, which is an increase compared to the earlier publication from this same reference lab population in which it was reported that 1.09% of samples had this mutation [23]. The increase in C480F prevalence may be due to increased frequency of MBV use, although controlled studies would be needed to conclusively prove an association.

Maribavir was approved in 2021 for post-transplant CMV infection/disease that is refractory to treatment with or without genotypic resistance testing for other anti-CMV drugs [24]. Earlier studies described MBV UL97 antiviral resistance mutations, including F342Y and C480F which confer resistance to both MBV and GCV [25,26]. Analysis of samples collected in the phase 3 trial found that baseline resistance to GCV, cidofovir and foscarnet was present in most patients in both the MBV arm and the investigator assigned therapy (IAT) arm [27]. However, MBV UL97 resistance at baseline was detected in only 3% of IAT arm subjects and none of the MBV arm subjects. Posttreatment testing demonstrated that MBV resistance mutations were present in 26% of subjects in the MBV arm and none of the IAT arm subjects. As found in the data presented herein, the most common MBV resistance mutations were T409M, H411Y and C480F. Recent case studies demonstrate the clinical impact of MBV resistance, as well as the presence of dual resistance to MBV and GCV [28,29]. Future studies are needed to determine if the use of MBV for treatment of resistant or refractory CMV infection has resulted in an increase in MBV resistance mutations in MBV treatment naïve patients presenting with CMV viremia or disease.

This study is limited due to the lack of specific patient information and de-identification of samples prior to analysis. Therefore, key information such as the type and timing of transplant performed, level of immunosuppression and the CMV antiviral drug history (drug(s) used, dose, length of treatment) is not known. Eurofins Viracor’s reference laboratory caseload primarily consists of samples from transplant recipients with a relatively even distribution between SOT and HCT recipients. Occasional samples are received from patients congenitally infected with CMV. An additional limitation is that assay validation experiments demonstrated minority populations (e.g., mixed infections of resistant and wild-type virus) were not detected by the Sanger sequencing method used when present at a level of <30% of the majority population, and thus samples with a low level of drug resistance variants would not be reported as resistant.

In summary, antiviral drug resistance mutations to GCV and MBV were commonly identified in samples analyzed by UL97 and UL54 genotypic resistance testing. Resistance to GCV was more common than MBV, and a majority of MBV-resistant samples also had mutations conferring resistance to GCV, either due to a single mutation conferring cross-resistance to both drugs or due to additional GCV UL97 or UL54 mutations, suggesting dual resistance to MBV and GCV may be common and clinically relevant, as recently presented in case reports [28,29].

## Figures and Tables

**Table 1 viruses-17-00421-t001:** Drug resistance mutations identified in viral genes UL54 (ganciclovir) and UL97 (maribavir and ganciclovir) in 1459 patient samples tested.

UL54			UL97					
Ganciclovir			Ganciclovir			Maribavir		
Mutation(s)	No.	%	Mutation(s)	No.	%	Mutation(s)	No.	%
None detected	1397	95.75%	None detected	1106	75.81%	None detected	1338	91.71%
N408D	1	0.07%	F342Y	6	0.41%	F342Y	6	0.41%
F412C	1	0.07%	K359E	1	0.07%	T409M	35	2.40%
F412L	1	0.07%	K359N	6	0.41%	T409M, H411N	1	0.07%
F412S	2	0.14%	K359E, C592G	1	0.07%	T409M, H411L	1	0.07%
D413E	4	0.27%	K359N, C480F	1	0.07%	T409M, H411Y	6	0.41%
D413E, D588N	3	0.21%	E362D	4	0.27%	T409M, C480F	2	0.14%
P488R	1	0.07%	M460I	10	0.69%	H411L	3	0.21%
L501I	6	0.41%	M460I, A594S	1	0.07%	H411Y	30	2.06%
T503I	6	0.41%	M460I, C603W	2	0.14%	H411N/Y	1	0.07%
T503I, D413E	1	0.07%	M460V/I	1	0.07%	H411Y, C480F	1	0.07%
T503I, P522S	1	0.07%	M460V	11	0.75%	C480F	35	2.40%
K513N	2	0.14%	M460V, H520Q	1	0.07%			
K513R	3	0.21%	M460V, L595F	1	0.07%			
K513T	1	0.07%	M460V, C603W	2	0.14%			
I521T	1	0.07%	C480F	33	2.26%			
P522S	12	0.82%	C480F, A594V	2	0.14%			
del524	1	0.07%	C480F, C603W	2	0.14%			
L545S	3	0.21%	H520Q	13	0.89%			
L545W	3	0.21%	H520Q, A594V	1	0.07%			
L545S, D301N	1	0.07%	H520Q, L595S	1	0.07%			
Q578H	1	0.07%	A591V	4	0.27%			
E756K	1	0.07%	C592F	1	0.07%			
A834P, G841A, A987G	1	0.07%	C592G	15	1.03%			
A987G	5	0.34%	C592G, L595S	1	0.07%			
			C592G, C603W	2	0.14%			
			A594E	1	0.07%			
			A594P	5	0.34%			
			A594S	2	0.14%			
			A594S, L595S	1	0.07%			
			A594T	5	0.34%			
			A594T, L595W	1	0.07%			
			A594V	48	3.29%			
			A594V, L595F	1	0.07%			
			A594V, L595S	2	0.14%			
			A594V, C603W	1	0.07%			
			del594-598	1	0.07%			
			del594-603	2	0.14%			
			L595F	15	1.03%			
			L595F, del599-603	1	0.07%			
			L595F, C603W	2	0.14%			
			L595S	71	4.87%			
			L595W	6	0.41%			
			del595-599	1	0.07%			
			del595-603	1	0.07%			
			del596	1	0.07%			
			E596G	1	0.07%			
			del600-603	1	0.07%			
			del601-603	1	0.07%			
			C603R	1	0.07%			
			C603W	54	3.70%			
			C603W, K359E	1	0.07%			
			C603W, C607F	1	0.07%			
			C607F	2	0.14%			

**Table 2 viruses-17-00421-t002:** Maribavir and ganciclovir drug resistance in 1459 patient samples tested.

Anti-CMV Drug and Viral Gene	No. Positive for Resistance	% Positive for Resistance
Maribavir UL97	121	8.29%
Ganciclovir	379	25.98%
UL54 mutation(s) only	26	1.78%
UL97 mutation(s) only	317	21.73%
Both UL54 and UL97 mutations	36	2.47%
Maribavir and ganciclovir	84	5.76%

## Data Availability

Data will be made available on request.

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
