# Peer review of "Dual Resistance to Maribavir and Ganciclovir in Transplant Recipients"

_viruses, 2025, doi:10.3390/v17030421_

Round 1

Reviewer 1 Report

Comments and Suggestions for Authors

This was a very interesting study describing the rates of CMV resistance testing since introducing maribavir to the market. I have a few comments for the authors:

1) Please consider introducing the concept of dual resistance mutation to maribavir and ganciclovir in the introduction (paragraph 2?). 

2) Is the author able to postulate by C480F mutation may be increasing? Is this because it arises after exposure to maribavir? Can we postulate that as maribavir is made more easily available, this mutation may increase with time?

3) Consider updating reference 5 with the updated definitions https://doi.org/10.1093/cid/ciae321 as the new definition for resistance and refractory is more clinically relevant. 

Author Response

Comment 1: Please consider introducing the concept of dual resistance mutation to maribavir and ganciclovir in the introduction (paragraph 2?).

Response 1: Thank you for this comment. I agree and have therefore added the following sentences to the introduction: “Patient samples with documented resistance mutations are most commonly resistant to a single CMV antiviral drug, although resistance to more than one drug has been reported.1,23 Resistance to multiple CMV antiviral drugs (e.g. dual-resistance) may be due to a single mutation conferring resistance to more than one drug or to the presence of multiple resistance mutations, each conferring resistance to different antiviral drugs.“ See page 2, paragraph 1, lines 46 – 50.

Comment 2: Is the author able to postulate by C480F mutation may be increasing? Is this because it arises after exposure to maribavir? Can we postulate that as maribavir is made more easily available, this mutation may increase with time?

Response 2: Thank you for this comment. I agree that it is reasonable to postulate that and have therefore added the following sentence: “The increase in C480F prevalence may be due to increased frequency of MBV use, although controlled studies would be needed to conclusively prove an association.” See page 6, paragraph 5, lines 150 – 152.

Comment 3: Consider updating reference 5 with the updated definitions https://doi.org/10.1093/cid/ciae321 as the new definition for resistance and refractory is more clinically relevant. 

Response 3: Thank you for this suggestion. I have updated reference 5 with the indicated manuscript. See page 8, lines 209 – 213.

Reviewer 2 Report

Comments and Suggestions for Authors

I think that the authors indicated valuable data in that it shows where the mutations are located in UL97 or UL54 and their frequency in samples from SOT and HSCT patients suspected of drug-resistant CMV infection. Since there is no clinical information at all, however, it is difficult to judge whether the frequency is high or low in samples from cases suspected of drug-resistant CMV infection. There is no information on the type and timing of transplant performed, level of immunosuppression and the CMV antiviral drug history (drug(s) used, dose, length of treatment), etc., so the clinical significance of CMV infection in those cases is unknown. Therefore, it is difficult to evaluate the data in this paper itself is important.

Could you please add any clinical information as many as possible?

Author Response

Comment 1: I think that the authors indicated valuable data in that it shows where the mutations are located in UL97 or UL54 and their frequency in samples from SOT and HSCT patients suspected of drug-resistant CMV infection. Since there is no clinical information at all, however, it is difficult to judge whether the frequency is high or low in samples from cases suspected of drug-resistant CMV infection. There is no information on the type and timing of transplant performed, level of immunosuppression and the CMV antiviral drug history (drug(s) used, dose, length of treatment), etc., so the clinical significance of CMV infection in those cases is unknown. Therefore, it is difficult to evaluate the data in this paper itself is important.

Could you please add any clinical information as many as possible?

Response 1: Thank you for this comment. I certainly agree that adding clinical information would be valuable, however a requirement of this analysis was that all samples be fully de-identified. No clinical information (CMV disease, immunosuppression, CMV antiviral drug history) was collected as part of the sample submission process to the Eurofins Viracor reference laboratory. Therefore, there is no ability to access clinical information retrospectively.

Reviewer 3 Report

Comments and Suggestions for Authors

The work of Dr Steven B. Kleiboeker addresses an important issue in the clinical application of chemotherapeutics for the treatment of patients with cytomegalovirus infection. This includes immunocompromised patients with solid organ transplantation (SOT) and allogeneic hematopoetic cell tranplantation (HTC). Standard therapy is carried out with ganciclovir (or the prodrug valganciclovir). The author also includes studies on the recently proven anti-cytomegalovirus drug maribavir. A precise genetic analysis of the characteristic mutations conferring drug resistance has been developed. A high value of the study is that it is based on 1,459 samples from patients with proven CMV. It is established that the use of maribavir is to a large extent an alternative to ganciclovir in terms of resistance. Methodologically, the article is based on precise methods for sequencing and analysis of the viral genome.
The work is a contribution to the treatment of cytomegalovirus infection.

Note: statistical analysis is missing.

Author Response

Reviewer 3

Comment 1: The work of Dr Steven B. Kleiboeker addresses an important issue in the clinical application of chemotherapeutics for the treatment of patients with cytomegalovirus infection. This includes immunocompromised patients with solid organ transplantation (SOT) and allogeneic hematopoetic cell tranplantation (HTC). Standard therapy is carried out with ganciclovir (or the prodrug valganciclovir). The author also includes studies on the recently proven anti-cytomegalovirus drug maribavir. A precise genetic analysis of the characteristic mutations conferring drug resistance has been developed. A high value of the study is that it is based on 1,459 samples from patients with proven CMV. It is established that the use of maribavir is to a large extent an alternative to ganciclovir in terms of resistance. Methodologically, the article is based on precise methods for sequencing and analysis of the viral genome. The work is a contribution to the treatment of cytomegalovirus infection.
Note: statistical analysis is missing.

Response 1: Thank you for this suggestion. The following statistical analysis and sentence was added: “Fisher’s exact test was applied to assess an association between GCV and MBV resistance and it was found to be highly significant (P <0.001).”  See page 6, paragraph 2, lines 123 – 124.

Round 2

Reviewer 2 Report

Comments and Suggestions for Authors

I don't have any particular opinion.

This paper is informative for readers.